evolution/biomechanics

body segment parameters, convex hulls, multibody dynamic analysis, Mammalia, biomechanics

**Author for correspondence:**
Samuel J. Coatham
e-mail: sam.coatham@postgrad.manchester.ac.uk

# Convex hull estimation of mammalian body segment parameters

Samuel J. Coatham[1], William I. Sellers[1] and Thomas A. Püschel[2,3]

[1]Faculty of Life Sciences, University of Manchester, Michael Smith Building, Oxford Road, Manchester M13 9PT, UK
[2]School of Biological Sciences, University of Reading, Reading RG6 6BX, UK
[3]Institute of Cognitive and Evolutionary Anthropology, University of Oxford, 64 Banbury Road, Oxford OX2 6PN, UK

SJC, 0000-0003-4597-6210; WIS, 0000-0002-2913-5406;
TAP, 0000-0002-2231-2297

Obtaining accurate values for body segment parameters (BSPs) is fundamental in many biomechanical studies, particularly for gait analysis. Convex hulling, where the smallest-possible convex object that surrounds a set of points is calculated, has been suggested as an effective and time-efficient method to estimate these parameters in extinct animals, where soft tissues are rarely preserved. We investigated the effectiveness of convex hull BSP estimation in a range of extant mammals, to inform the potential future usage of this technique with extinct taxa. Computed tomography scans of both the skeleton and skin of every species investigated were virtually segmented. BSPs (the mass, position of the centre of mass and inertial tensors of each segment) were calculated from the resultant soft tissue segments, while the bone segments were used as the basis for convex hull reconstructions. We performed phylogenetic generalized least squares and ordinary least squares regressions to compare the BSPs calculated from soft tissue segments with those estimated using convex hulls, finding consistent predictive relationships for each body segment. The resultant regression equations can, therefore, be used with confidence in future volumetric reconstruction and biomechanical analyses of mammals, in both extinct and extant species where such data may not be available.

## 1. Introduction

Accurate volumetric reconstruction of extinct taxa is an important tool in palaeobiology. Soft tissue reconstruction has various research functions [1], while body mass has a range of biological correlates [2,3]. This is especially true in biomechanical studies

using techniques such as multibody dynamics analysis (MDA) [4], where it is essential to estimate the specific mass and inertial properties for each body segment [5]. Body segment parameters (BSPs) comprise the mass, position of the centre of mass and the inertial tensor of a segment. Accurate BSP estimation is fundamental to biomechanics [6], having been repeatedly shown, as would be expected, to significantly impact the results of MDA [7–11].

In human subjects, BSPs have historically been estimated based on calibration studies of human cadavers [12,13], with the resultant regression equations used to calculate parameter values in living subjects [6]. More recently, three-dimensional imaging techniques—including X-ray, computed tomography (CT) and photogrammetry—have been used to calculate BSPs in living individuals [11,14–16]. However, the scarcity of fossils with soft tissue preservation prevents equivalent calculation of BSPs in extinct animals [17].

Many techniques for volumetric reconstruction based on skeletal data have been proposed, including elliptical hooping [18,19] and the usage of B-spline solids [5]. Non-uniform rational B-splines (NURBS) have been used to estimate BSPs of extinct taxa [3], yielding promising results [20,21]. However, NURBS reconstruction still requires a degree of user input [2] for the manual creation of soft tissues [19]. Convex hull reconstruction, in which all the points of a functional segment are 'shrink-wrapped' by a three-dimensional shape, has been suggested as an alternative [17]. It is extremely quick and mostly objective, assuming accurate re-articulation of the fossil specimen [2]. While the resultant shape is unlikely to visually match the original segment, convex hulling has seen widespread adoption in recent years [22–28]. This has only occasionally been extended to BSP estimation for subsequent gait analyses [29,30], and the accuracy of this approach is currently unknown.

For accurate convex hull reconstruction, a validatory dataset must first be compiled, made up of extant species related to the specimen under investigation [2]. The naive convex hulls can subsequently be converted using calibrated models based on this dataset, to replicate the relationship observed between convex hull reconstructions of extant taxa and their true mass properties [31]. In this study, we have built a dataset of extant mammals, to inform future mammal reconstructions using convex hulls. This can dramatically increase the speed with which convex hull reconstructions of mammals are performed in future, as our predictive models can be incorporated quickly and easily, without the need to produce an equivalent validatory dataset.

# 2. Material and methods

A dataset of 32 mammal species (spanning nine orders) was produced for this study, with species selected based on the availability of CT scan data, with the goal of providing some degree of phylogenetic coverage across Mammalia (figure 1). Body masses ranged from around 20 g in *Mus musculus* to over 400 kg in *Ursus maritimus* [33]. CT scans of living or recently deceased individuals were provided by a combination of universities, museums, imaging companies and zoos (electronic supplementary material, table S1). Scans comprising the entire body of an individual were preferred; this was frequently not possible, for a variety of reasons. Often, the specimen had only been scanned partially, either due to size or because specific regions were targeted for medical uses. In other cases, the condition of certain segments was deemed unsuitable for accurate volumetric reconstruction, for example, the chest cavities of multiple animals had been emptied during autopsy. The issue of damage to the torso is well known in BSP studies [24,34].

Raw CT data for each specimen was processed using RadiAnt DICOM viewer (Radiant DICOM viewer 2020.2, Medixant, Poznań, Poland), resulting in three-dimensional surfaces of both the skin and skeleton of each individual. These were imported into Geomagic Studio (Geomagic Studio® 2013, 3D Systems Inc., Rock Hill, SC, USA), where the surfaces were manually 'cleaned', removing artificial objects like collars, then segmented. Segmentation involved subdividing the body into functional segments, as has been previously established for convex hull reconstructions [31] and BSP calculations [6,34]: three segments for each limb, while the axial skeleton was split into skull, neck, trunk and tail. Soft tissue segments were divided based on the underlying skeletal segments, in a manner roughly analogous to that used to calculate BSPs from cadavers [13,35]. Apart from the tail segments, all the rest were separated at the locations of joints. For example, the head was separated from the neck at the atlanto-occipital joint and the neck was separated from the trunk at the cervicothoracic joint. This segmentation could be performed with very high precision with the skeleton, while there was a degree of user interpretation necessary for segmentation of the skin where joint locations were less clear. The basic workflow for preparing each specimen is illustrated in figure 2.

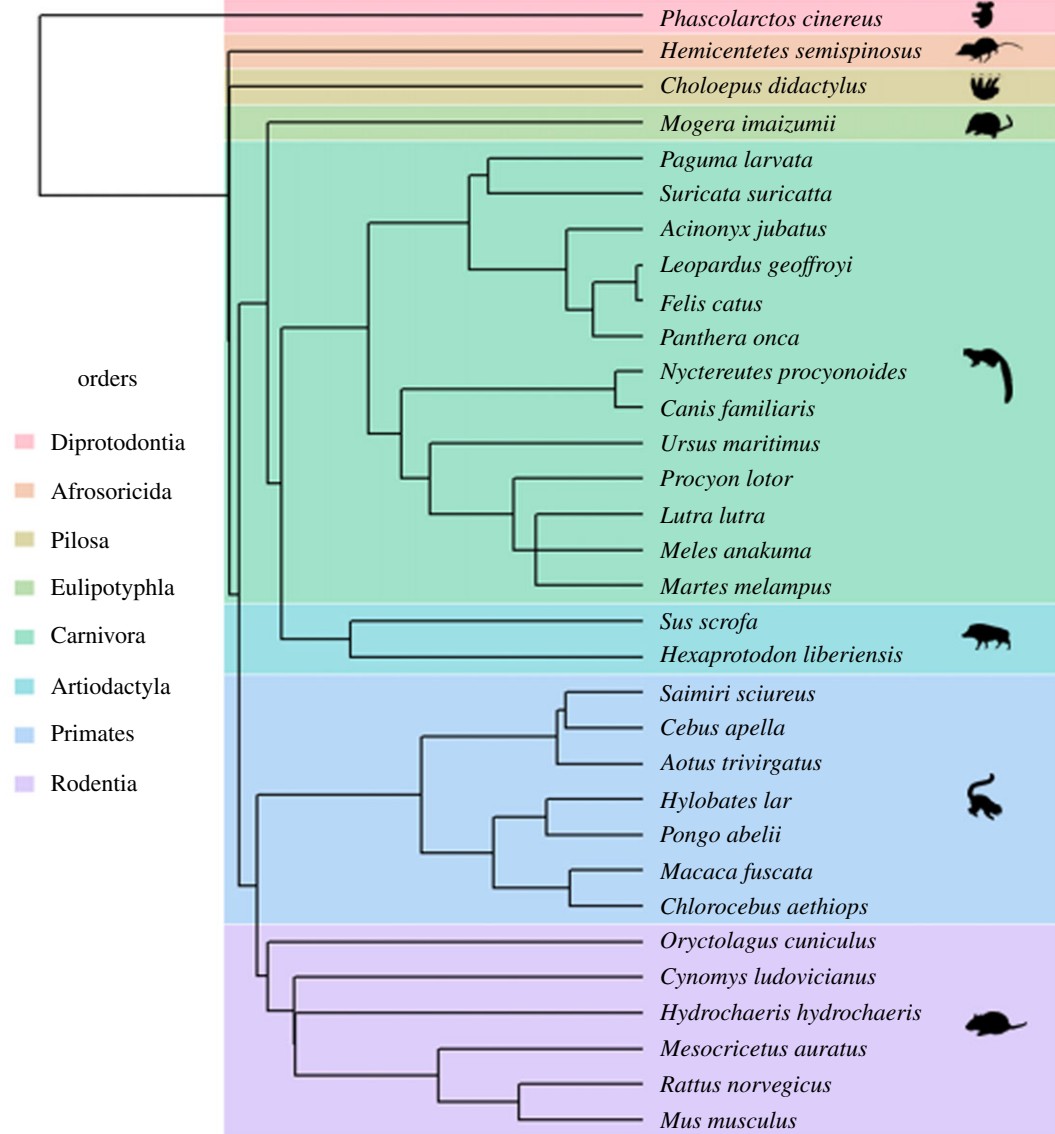

**Figure 1.** Simplified phylogeny showing the species used in this study, derived from the mammalian supertree in the R *treeman* package [32]. Animal outlines are taken from PhyloPic, under the *Public Domain Dedication 1.0* license. The Afrosoricida outline was produced by Mo Hassan, under the *Creative Commons Attribution-NonCommercial-ShareAlike 3.0 Unported* license.

Curvature in the tails of some specimens would have produced unrealistic convex hull reconstructions, and manually straightening these specimens in three-dimensional modelling software would have prevented accurate calculation of mass properties from the resultant soft tissue segment. Consequently, the tails of all individuals were separated into four roughly equal segments, to enable the calculation of more lifelike convex hulls. Subdividing a segment for more accurate convex hull reconstruction has precedent, with long necks having been similarly split in previous work [24,31]. The position of the scanned animals had some impact on the position of the skin. For example, some larger specimens were restrained to ensure their fit in a CT scanner, resulting in unnatural bulging in some segments that may have altered their shape (and consequently BSPs). Segments that were clearly substantially impacted in this manner were removed from the analysis, while minor instances were manually repaired (in the form of cleaning and smoothing) where it was deemed necessary. Multiple specimens were in a curled position with the limbs tucked tightly into the torso, resulting in a warped torso shape that was difficult to rectify without considerable subjective alteration. Ideally, this would be avoided by reposing the thawed cadaver prior to CT scanning; however, this was not a practical option for this study.

The skin and underlying skeletal segments were imported into Blender (Blender 2.81, Blender Foundation, Amsterdam, The Netherlands), where they were repositioned so that all limb joints were in a columnar pose [36]. Convex hulls based on the skeletal segments were produced in MeshLab

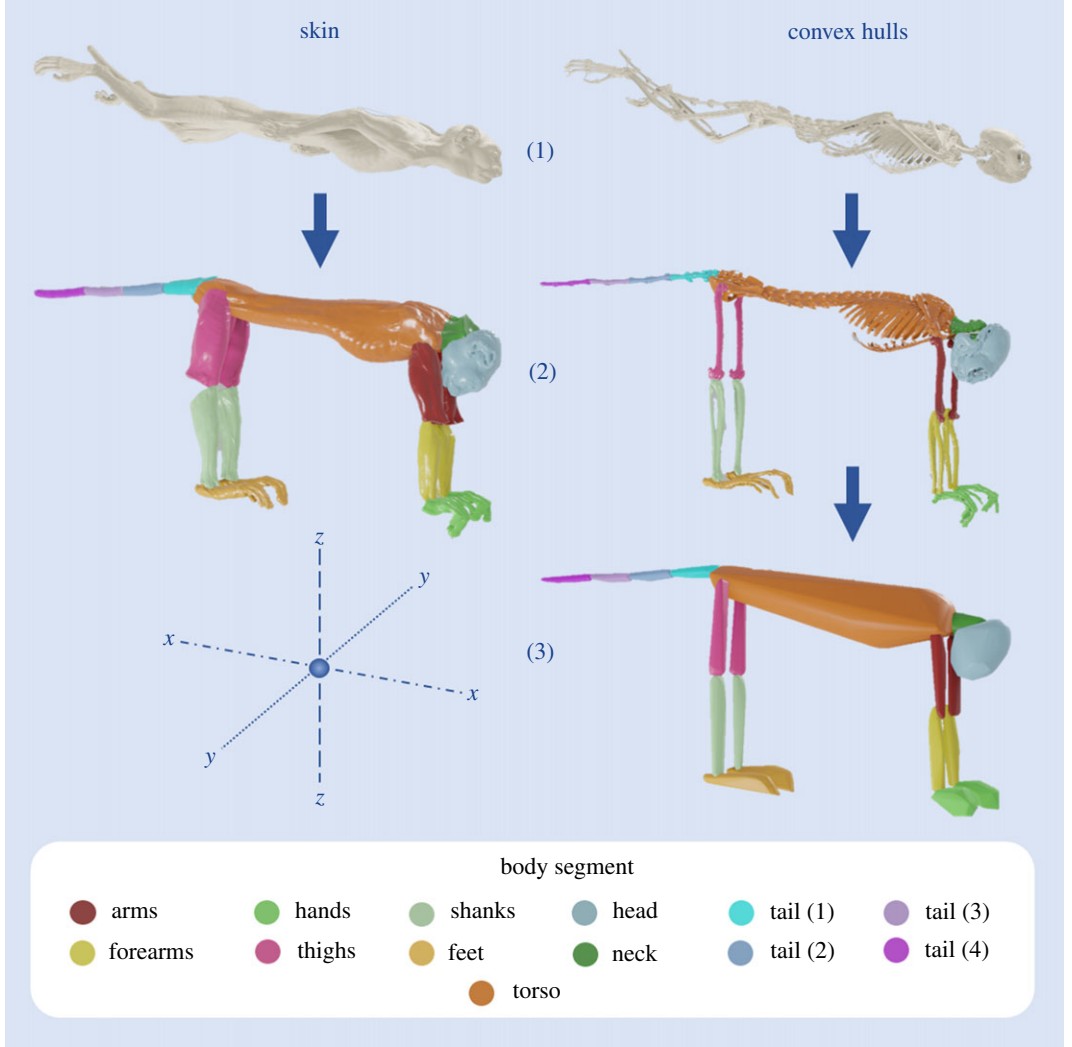

**Figure 2.** Basic workflow for preparing skin and convex hull segments from whole-body CT scans. Initial skin and skeletal CT scans of a three-striped night monkey (*Aotus trivirgatus*). The scans are segmented and re-posed in a columnar posture. Convex hulls are calculated for each skeletal segment. The axes are displayed, with *x* referring to the craniocaudal, *y* referring to the mediolateral axis and *z* referring to the vertical axis.

(MeshLab 2020.12, Visual Computing Lab, ISTI-CNR) [37], computed using the Quickhull algorithm [38]. BSPs of the convex hull and soft tissue segments were then calculated in GaitSym (GaitSym 2019, https://github.com/wol101/GaitSym2019) [29], which implements the computational methods outlined in Mirtich [39]. GaitSym was selected for the ease of outputting the BSP data, although MeshLab is also capable of calculating BSPs (and produced identical results).

The 10 BSPs assessed in this experiment were

(1) The mass of the segment. In this study, a nominal density of 1000 kg m$^{-3}$ was used, but subject-specific (and segment-specific) density values should be used where possible.

(2–4) The coordinates of the centre of mass of the segment. CM (*x*), CM (*y*) and CM (*z*) define the location of the segmental centre of mass in the *x* (craniocaudal), *y* (mediolateral) and *z* (vertical) axes, respectively.

(5–10) The inertial tensors of the segment. There are three moments of inertia, *Ixx*, *Iyy* and *Izz*. These define the resistance to rotation in axes *x*, *y* and *z*, respectively, when torque is applied about that specific axis—higher values mean that greater force is required in order for the segment to rotate in that axis. Similarly, the products of inertia *Ixy*, *Ixz* and *Iyz*, define the resistance to rotation in one axis when torque is applied in a different axis. For example, *Ixy* can be understood as the resistance to rotation in the *x*-axis when torque is applied in the *y*-axis. For all six inertial tensors, rotational resistance is determined by the distribution of mass in a segment around the rotational axis.

The potential relationships between BSPs estimated based on the convex hull reconstructions and those calculated from soft tissue segments were investigated, for every combination of body segment and BSP. Axial segments were laterally symmetrical in the $y$-axis, with the centre of mass positioned at $y = 0$ (although this was not always the case in practice, due to the imperfect nature of the scanning and re-positioning process). Consequently, CM ($y$) as well as the products of inertia $Ixy$ and $Iyz$ should always be set as equal to zero in MDA, although the results for these BSPs are still reported in this study. All values for mass and the three moments of inertia were log-transformed ($\log_{10}$) prior to the regressions. This improved the predictive effectiveness of the resultant equations and accounted for the impact of scaling on the regressions.

In total, 130 regressions were performed. Ordinary least squares (OLS) regressions were carried out in each instance, in line with previous work [25]. Phylogenetic generalized least squares (PGLS) regressions were performed concurrently [40], to account for any potential phylogenetic signal in the dataset. The phylogeny used for this study was a modified version of the mammalian supertree produced by Bininda-Emonds *et al.* [32] (figure 1). Empirically, PGLS regressions optimizing lambda transformations using maximum likelihood revealed that, for most segments, the sample size was insufficient to confidently estimate a value for lambda [41]. Instead, lambda was fixed at 1 for the PGLS regressions—assuming Brownian motion [42]. This was contrasted with the OLS regressions (equivalent to lambda being fixed at 0), with the best-fitting model being selected, using the Akaike information criterion (AIC) [43]. The validity of the OLS regressions were further assessed using leave-one-out cross-validation (LOOCV) [44,45]. This was selected as it is optimal for use with smaller samples [46]. The root mean square error of prediction (RMSE) was calculated in each instance [47].

The resultant regression equations for every BSP-segment combination can serve as predictive models, with which BSPs can be approximated based on convex hull values. This method has been used extensively in mass reconstructions using convex hulls, typically to produce a single regression equation explaining the relationship between cumulative convex hull volume and total body mass [17,24,31]. Following the example of a prior convex hull experiment [25], $R^2$ values were reported for OLS regressions but not PGLS regressions, due to the differing definitions of $R^2$ between the two operations [40].

For the purposes of this study, the appendicular segments from the left side of the body were used to produce the regression equations for both left and right segments, to ensure consistency in reconstruction of the limbs. This was achieved by simply reversing the signs for CM ($y$), $Ixy$ and $Iyz$ for the equations of all right-sided appendicular segments. The decision to use left-sided segments instead of right as the basis was arbitrary, as both sides had the same number of samples and produced similar results. The mean difference between the slopes of left and right limb segments was 1.3%, indicating that this approach had little impact on the accuracy of the results.

A density value of 1000 kg m$^{-3}$, the density of water, was applied to all segments. This has been frequently used in previous reconstruction work [3,5,48,49]. Where possible, subject-specific density values should be applied in reconstruction—calculated directly when researching extant taxa, or approximated using the extant phylogenetic bracket [1] for extinct taxa. This is likely to improve the accuracy of estimation [5]. Experimentally derived density values can easily be applied with the method outlined here, as BSPs scale with density (assuming uniform density across a segment). For example, with a desired density of 900 kg m$^{-3}$, reconstructed mass and inertial tensor values would simply need to be multiplied by 0.9—with the coordinates of the centre of mass staying the same.

## 3. Results

In general, OLS regressions fit the data better than PGLS regressions. When comparing the two, OLS regressions resulted in lower AIC values in 110 out of the total 130 regressions performed. Consequently, the results of the OLS regressions will be primarily discussed throughout this study. All PGLS results are also made available in the electronic supplementary material. This is consistent with a previous study [24], which used similar methodology and also found OLS regressions were better-fitting for their data.

The calculated OLS regression equations for each segment are shown in table 1, with confidence intervals for the intercept and slope of each regression displayed in tables 2 and 3, respectively. Expanded results are available in electronic supplementary material, tables S2–S14. Overall, the investigation revealed highly significant ($p < 0.01$) predictive relationships between BSPs estimated using convex hulls and BSPs calculated from soft tissue for every segment–parameter combination tested. Most of the regressions performed had $R^2$ values greater than 0.9, representing 116 out of 130 relationships (89%)—the $R^2$ distribution is illustrated in electronic supplementary material, figure S1.

**Table 1.** OLS regression equations for every segment–parameter relationship, to convert the naive convex hull-estimated parameter value to a more realistic value. Appendicular equations are derived from segments on the left side of the body. When reconstructing right limb segments, the same equations can be used—simply reverse the signs for CM ($y$), $I_{xy}$ and $I_{yz}$ (for example, $1.55 \times 10^{-3} + 0.97\times$ becomes $-1.55 \times 10^{-3} + 0.97\times$). Calculated equations for CM ($y$), $I_{xy}$ and $I_{yz}$ in the axial segments are included, but these values can be set to zero in MDA.

| body segment | $\log_{10}$ mass (kg) | CM ($x$) (m) | CM ($y$) (m) | CM ($z$) (m) | $\log_{10}$ $I_{xx}$ (kg m²) | $\log_{10}$ $I_{yy}$ (kg m²) | $\log_{10}$ $I_{zz}$ (kg m²) | $I_{xy}$ (kg m²) | $I_{xz}$ (kg m²) | $I_{yz}$ (kg m²) |
|---|---|---|---|---|---|---|---|---|---|---|
| left arm | $3.05 \times 10^{-1} + 0.88x$ | $4.45 \times 10^{-4} + 0.99x$ | $1.01 \times 10^{-3} + 0.98x$ | $-5.82 \times 10^{-4} + 1.01x$ | $2.60 \times 10^{-1} + 0.93x$ | $3.04 \times 10^{-1} + 0.93x$ | $2.91 \times 10^{-1} + 0.92x$ | $-2.55 \times 10^{-3} + 1.83x$ | $-1.43 \times 10^{-2} + 1.85x$ | $-1.23 \times 10^{-3} + 1.84x$ |
| left forearm | $2.44 \times 10^{-1} + 0.91x$ | $-3.96 \times 10^{-7} + 1.00x$ | $-1.33 \times 10^{-3} + 1.01x$ | $-7.07 \times 10^{-4} + 1.08x$ | $2.43 \times 10^{-1} + 0.95x$ | $2.69 \times 10^{-1} + 0.95x$ | $2.62 \times 10^{-1} + 0.95x$ | $8.19 \times 10^{-4} + 2.24x$ | $8.08 \times 10^{-4} + 2.34x$ | $-3.05 \times 10^{-6} + 2.07x$ |
| left hand | $-9.11 \times 10^{-2} + 0.95x$ | $-1.25 \times 10^{-4} + 0.99x$ | $6.57 \times 10^{-4} + 0.99x$ | $9.64 \times 10^{-4} + 0.98x$ | $-1.40 \times 10^{-1} + 0.97x$ | $-9.20 \times 10^{-2} + 0.97x$ | $-9.14 \times 10^{-2} + 0.97x$ | $-1.30 \times 10^{-4} + 0.77x$ | $-8.20 \times 10^{-5} + 0.84x$ | $-7.49 \times 10^{-6} + 0.87x$ |
| left thigh | $5.49 \times 10^{-1} + 0.94x$ | $-1.86 \times 10^{-3} + 0.87x$ | $2.74 \times 10^{-4} + 1.04x$ | $-1.95 \times 10^{-3} + 1.05x$ | $5.72 \times 10^{-1} + 0.97x$ | $5.76 \times 10^{-1} + 0.97x$ | $4.82 \times 10^{-1} + 0.95x$ | $2.33 \times 10^{-5} + 3.84x$ | $7.82 \times 10^{-4} + 4.52x$ | $-1.29 \times 10^{-3} + 3.77x$ |
| left shank | $1.74 \times 10^{-1} + 0.92x$ | $-1.47 \times 10^{-3} + 0.90x$ | $7.55 \times 10^{-4} + 1.02x$ | $-1.75 \times 10^{-3} + 1.04x$ | $1.32 \times 10^{-1} + 0.95x$ | $1.19 \times 10^{-1} + 0.95x$ | $8.12 \times 10^{-2} + 0.94x$ | $1.52 \times 10^{-5} + 1.54x$ | $2.21 \times 10^{-5} + 1.50x$ | $-6.29 \times 10^{-5} + 1.51x$ |
| left foot | $-6.52 \times 10^{-3} + 0.99x$ | $-2.90 \times 10^{-3} + 0.99x$ | $-5.62 \times 10^{-5} + 1.01x$ | $1.24 \times 10^{-3} + 0.97x$ | $-1.52 \times 10^{-3} + 0.99x$ | $-2.83 \times 10^{-2} + 1.00x$ | $-1.74 \times 10^{-2} + 1.00x$ | $-1.76 \times 10^{-5} + 0.87x$ | $1.26 \times 10^{-6} + 1.68x$ | $6.71 \times 10^{-6} + 1.85x$ |
| head | $1.06 \times 10^{-1} + 0.93x$ | $8.99 \times 10^{-5} + 1.00x$ | $-2.12 \times 10^{-4} + 1.25x$ | $3.96 \times 10^{-3} + 0.99x$ | $6.40 \times 10^{-2} + 0.95x$ | $9.28 \times 10^{-2} + 0.96x$ | $8.89 \times 10^{-2} + 0.96x$ | $-5.84 \times 10^{-5} + 2.45x$ | $4.08 \times 10^{-3} + 1.43x$ | $-2.62 \times 10^{-5} + 2.52x$ |
| neck | $4.77 \times 10^{-1} + 0.87x$ | $4.51 \times 10^{-4} + 1.00x$ | $-1.96 \times 10^{-3} + 1.01x$ | $-1.66 \times 10^{-3} + 1.01x$ | $5.25 \times 10^{-1} + 0.95x$ | $5.14 \times 10^{-1} + 0.93x$ | $4.96 \times 10^{-1} + 0.93x$ | $3.83 \times 10^{-4} + 8.13x$ | $-2.78 \times 10^{-2} + 3.44x$ | $2.32 \times 10^{-4} + 7.96x$ |
| torso | $1.61 \times 10^{-1} + 0.94x$ | $2.09 \times 10^{-3} + 0.93x$ | $4.66 \times 10^{-4} + 0.58x$ | $4.21 \times 10^{-4} + 0.98x$ | $9.29 \times 10^{-2} + 0.96x$ | $1.00 \times 10^{-1} + 0.97x$ | $8.80 \times 10^{-2} + 0.97x$ | $-2.28 \times 10^{-3} + 0.67x$ | $-7.49 \times 10^{-2} + 1.18x$ | $-3.12 \times 10^{-3} + 0.35x$ |
| tail 1 | $2.94 \times 10^{-1} + 0.89x$ | $1.16 \times 10^{-4} + 0.98x$ | $7.10 \times 10^{-4} + 0.84x$ | $-1.19 \times 10^{-3} + 1.01x$ | $3.57 \times 10^{-1} + 0.94x$ | $3.51 \times 10^{-1} + 0.94x$ | $1.42 \times 10^{-1} + 0.92x$ | $1.66 \times 10^{-6} + 7.68x$ | $3.23 \times 10^{-5} + 2.06x$ | $-3.10 \times 10^{-6} + 6.58x$ |
| tail 2 | $1.03 \times 10^{-1} + 0.87x$ | $9.40 \times 10^{-4} + 1.00x$ | $3.98 \times 10^{-4} + 0.91x$ | $-2.59 \times 10^{-4} + 1.00x$ | $2.09 \times 10^{-1} + 0.94x$ | $1.95 \times 10^{-1} + 0.94x$ | $1.03 \times 10^{-2} + 0.92x$ | $1.05 \times 10^{-6} + 1.85x$ | $1.17 \times 10^{-4} + 1.76x$ | $-6.46 \times 10^{-6} + 1.89x$ |
| tail 3 | $-1.39 \times 10^{-2} + 0.88x$ | $-2.71 \times 10^{-3} + 0.98x$ | $8.55 \times 10^{-5} + 1.27x$ | $-6.91 \times 10^{-4} + 1.01x$ | $1.35 \times 10^{-1} + 0.95x$ | $9.76 \times 10^{-2} + 0.95x$ | $-6.26 \times 10^{-2} + 0.92x$ | $4.52 \times 10^{-7} + 1.82x$ | $1.09 \times 10^{-4} + 1.08x$ | $-3.60 \times 10^{-7} + 1.85x$ |
| tail 4 | $-3.40 \times 10^{-2} + 0.87x$ | $-5.52 \times 10^{-3} + 0.98x$ | $-1.11 \times 10^{-4} + 1.16x$ | $1.29 \times 10^{-4} + 1.00x$ | $1.05 \times 10^{-1} + 0.94x$ | $7.00 \times 10^{-2} + 0.93x$ | $-2.14 \times 10^{-2} + 0.92x$ | $1.25 \times 10^{-7} + 1.19x$ | $1.09 \times 10^{-4} + 1.00x$ | $-2.40 \times 10^{-7} + 1.17x$ |

**Table 2.** 95% confidence intervals of the intercept for every segment–parameter relationship, derived from OLS regressions. (a) (±95%).

| body segment | $\log_{10}$ mass (kg) | CM (x) (m) | CM (y) (m) | CM (z) (m) | $\log_{10}$ Ixx (kg m²) | $\log_{10}$ Iyy (kg m²) | $\log_{10}$ Izz (kg m²) | Ixy (kg m²) | Ixz (kg m²) | Iyz (kg m²) |
|---|---|---|---|---|---|---|---|---|---|---|
| left arm | 0.1833–0.4264 | −0.0023–0.0032 | −0.0007–0.0027 | −0.0048–0.0037 | 0.1344–0.3864 | 0.1854–0.4229 | 0.1633–0.4178 | −0.0060–0.0009 | −0.0368–0.0083 | −0.0029–0.0005 |
| left forearm | 0.1332–0.3539 | −0.0010–0.0010 | −0.0031–0.0005 | −0.0037–0.0022 | 0.0965–0.3901 | 0.1525–0.3848 | 0.1433–0.3811 | −0.0024–0.0040 | −0.0063–0.0079 | −0.0013–0.0013 |
| left hand | −0.2097–0.0274 | $-2.27 \times 10^{-3}$–$2.02 \times 10^{-3}$ | $6.29 \times 10^{-5}$–$1.25 \times 10^{-3}$ | $7.36 \times 10^{-6}$–$1.92 \times 10^{-3}$ | −0.3217–0.0421 | −0.2115–0.0276 | −0.2107–0.0279 | $-3.68 \times 10^{-4}$–$1.07 \times 10^{-4}$ | $-1.61 \times 10^{-4}$–$2.57 \times 10^{-6}$ | $-1.41 \times 10^{-5}$–$8.58 \times 10^{-7}$ |
| left thigh | 0.3959–0.7023 | $-5.08 \times 10^{-3}$–$1.37 \times 10^{-3}$ | $-1.87 \times 10^{-3}$–$2.42 \times 10^{-3}$ | $-6.67 \times 10^{-3}$–$2.76 \times 10^{-3}$ | 0.4006–0.7438 | 0.4039–0.7489 | 0.2034–0.7604 | $-3.68 \times 10^{-4}$–$4.14 \times 10^{-4}$ | $-7.30 \times 10^{-4}$–$2.29 \times 10^{-3}$ | $-6.43 \times 10^{-3}$–$3.84 \times 10^{-3}$ |
| left shank | 0.0626–0.2851 | $-3.45 \times 10^{-3}$–$5.00 \times 10^{-4}$ | $-3.75 \times 10^{-4}$–$1.89 \times 10^{-3}$ | $-5.27 \times 10^{-3}$–$1.77 \times 10^{-3}$ | −0.0081–0.2726 | −0.0179–0.2569 | −0.0917–0.2541 | $-4.95 \times 10^{-5}$–$7.99 \times 10^{-5}$ | $-5.98 \times 10^{-5}$–$1.04 \times 10^{-4}$ | $-3.65 \times 10^{-4}$–$2.39 \times 10^{-4}$ |
| left foot | −0.1069–0.0939 | $-5.25 \times 10^{-3}$–$5.48 \times 10^{-4}$ | $-4.62 \times 10^{-4}$–$3.50 \times 10^{-4}$ | $2.70 \times 10^{-4}$–$2.21 \times 10^{-3}$ | −0.1567–0.1537 | −0.1970–0.1404 | −0.1680–0.1331 | $-1.05 \times 10^{-4}$–$7.02 \times 10^{-5}$ | $-8.43 \times 10^{-6}$–$1.10 \times 10^{-5}$ | $-6.19 \times 10^{-6}$–$1.96 \times 10^{-5}$ |
| head | 0.0433–0.1679 | $-2.39 \times 10^{-3}$–$2.57 \times 10^{-3}$ | $-8.08 \times 10^{-4}$–$3.84 \times 10^{-4}$ | $2.70 \times 10^{-4}$–$7.66 \times 10^{-3}$ | −0.0360–0.1640 | 0.0260–0.1595 | 0.0209–0.1569 | $-4.43 \times 10^{-4}$–$3.26 \times 10^{-4}$ | $-3.05 \times 10^{-2}$–$3.86 \times 10^{-2}$ | $-2.23 \times 10^{-4}$–$1.70 \times 10^{-4}$ |
| neck | 0.2418–0.7122 | $-4.70 \times 10^{-3}$–$5.60 \times 10^{-3}$ | $-3.81 \times 10^{-3}$–$1.18 \times 10^{-4}$ | $-4.94 \times 10^{-3}$–$1.61 \times 10^{-3}$ | 0.2786–0.7705 | 0.2930–0.7350 | 0.2613–0.7313 | $-8.75 \times 10^{-4}$–$1.64 \times 10^{-3}$ | $-9.09 \times 10^{-2}$–$3.53 \times 10^{-2}$ | $-4.86 \times 10^{-4}$–$9.50 \times 10^{-4}$ |
| torso | 0.0803–0.2413 | $-4.53 \times 10^{-3}$–$8.72 \times 10^{-3}$ | $-7.07 \times 10^{-4}$–$1.64 \times 10^{-3}$ | $-6.03 \times 10^{-3}$–$6.87 \times 10^{-3}$ | −0.0112–0.1969 | 0.0130–0.1877 | −0.0038–0.1798 | $-6.93 \times 10^{-3}$–$2.36 \times 10^{-3}$ | $-3.63 \times 10^{-1}$–$2.13 \times 10^{-1}$ | $-9.97 \times 10^{-3}$–$3.74 \times 10^{-3}$ |
| tail 1 | −0.0215–0.6092 | $-2.74 \times 10^{-3}$–$2.97 \times 10^{-3}$ | $1.67 \times 10^{-4}$–$1.25 \times 10^{-3}$ | $-3.01 \times 10^{-3}$–$6.20 \times 10^{-4}$ | 0.0303–0.6828 | 0.0270–0.6757 | −0.3679–0.6510 | $-3.66 \times 10^{-7}$–$3.69 \times 10^{-6}$ | $-4.90 \times 10^{-6}$–$6.94 \times 10^{-5}$ | $-2.53 \times 10^{-5}$–$1.92 \times 10^{-5}$ |
| tail 2 | −0.2647–0.4706 | $-9.35 \times 10^{-4}$–$2.81 \times 10^{-3}$ | $1.07 \times 10^{-4}$–$6.89 \times 10^{-4}$ | $-1.16 \times 10^{-3}$–$6.44 \times 10^{-4}$ | −0.1830–0.6019 | −0.1911–0.5814 | −0.4103–0.4309 | $-1.41 \times 10^{-6}$–$3.51 \times 10^{-6}$ | $-1.33 \times 10^{-4}$–$3.67 \times 10^{-4}$ | $-2.55 \times 10^{-5}$–$1.26 \times 10^{-5}$ |
| tail 3 | −0.4393–0.4115 | $-6.24 \times 10^{-3}$–$8.13 \times 10^{-4}$ | $-4.06 \times 10^{-4}$–$5.77 \times 10^{-4}$ | $-2.20 \times 10^{-3}$–$8.22 \times 10^{-4}$ | −0.3082–0.5783 | −0.3282–0.5235 | −0.5266–0.4013 | $-3.75 \times 10^{-7}$–$1.28 \times 10^{-6}$ | $-8.33 \times 10^{-5}$–$3.01 \times 10^{-4}$ | $-1.54 \times 10^{-6}$–$8.24 \times 10^{-7}$ |
| tail 4 | −0.5004–0.4325 | $-1.07 \times 10^{-2}$–$-3.15 \times 10^{-4}$ | $-4.42 \times 10^{-4}$–$2.20 \times 10^{-4}$ | $-3.85 \times 10^{-4}$–$6.42 \times 10^{-4}$ | −0.3459–0.5562 | −0.3502–0.4902 | −0.4626–0.4197 | $-4.46 \times 10^{-4}$–$6.96 \times 10^{-7}$ | $-6.94 \times 10^{-5}$–$2.88 \times 10^{-4}$ | $-9.76 \times 10^{-7}$–$4.96 \times 10^{-7}$ |

**Table 3.** 95% confidence intervals of the slope for every segment–parameter relationship, derived from OLS regressions. (b) (±95%).

| body segment | $\log_{10}$ mass (kg) | CM (x) (m) | CM (y) (m) | CM (z) (m) | $\log_{10}$ Ixx (kg m²) | $\log_{10}$ Iyy (kg m²) | $\log_{10}$ Izz (kg m²) | Ixy (kg m²) | Ixz (kg m²) | Iyz (kg m²) |
|---|---|---|---|---|---|---|---|---|---|---|
| left arm | 0.82–0.94 | 0.98–1.00 | 0.95–1.02 | 0.99–1.03 | 0.89–0.96 | 0.89–0.96 | 0.89–0.96 | 1.70–1.95 | 1.67–2.02 | 1.71–1.97 |
| left forearm | 0.85–0.96 | 1.00–1.01 | 0.97–1.04 | 1.05–1.11 | 0.92–0.99 | 0.92–0.99 | 0.91–0.99 | 2.07–2.41 | 2.16–2.52 | 1.78–2.36 |
| left hand | 0.90–1.01 | 0.98–1.00 | 0.98–1.00 | 0.84–1.12 | 0.93–1.00 | 0.94–1.01 | 0.94–1.01 | 0.76–0.79 | 0.76–0.92 | 0.76–0.97 |
| left thigh | 0.86–1.02 | 0.76–0.99 | 1.00–1.09 | 1.03–1.07 | 0.92–1.02 | 0.92–1.02 | 0.89–1.01 | 3.37–4.32 | 4.13–4.90 | 2.89–4.65 |
| left shank | 0.86–0.98 | 0.82–0.99 | 1.00–1.04 | 1.00–1.08 | 0.92–0.98 | 0.92–0.98 | 0.91–0.98 | 1.43–1.66 | 1.43–1.57 | 1.36–1.67 |
| left foot | 0.94–1.04 | 0.95–1.03 | 1.00–1.02 | 0.81–1.12 | 0.96–1.03 | 0.97–1.04 | 0.97–1.03 | 0.81–0.93 | 1.51–1.84 | 1.71–1.99 |
| head | 0.88–0.99 | 0.99–1.00 | 0.83–1.66 | 0.98–1.01 | 0.92–0.99 | 0.93–0.99 | 0.93–0.99 | 2.26–2.63 | 1.28–1.57 | 2.24–2.80 |
| neck | 0.75–0.99 | 0.99–1.01 | 0.33–1.70 | 1.00–1.02 | 0.88–1.02 | 0.86–1.01 | 0.85–1.00 | 4.62–11.64 | 2.57–4.32 | 4.78–11.15 |
| torso | 0.85–1.02 | 0.90–0.96 | 0.25–0.90 | 0.96–1.01 | 0.92–1.00 | 0.93–1.00 | 0.93–1.00 | 0.54–0.81 | 1.04–1.33 | 0.16–0.54 |
| tail 1 | 0.78–1.00 | 0.87–1.09 | 0.67–1.01 | 1.00–1.02 | 0.87–1.02 | 0.87–1.02 | 0.84–1.00 | 6.82–8.53 | 2.03–2.08 | 5.84–7.33 |
| tail 2 | 0.75–0.99 | 0.98–1.02 | 0.79–1.03 | 1.00–1.01 | 0.86–1.02 | 0.85–1.02 | 0.84–0.99 | 1.68–2.03 | 1.71–1.81 | 1.41–2.37 |
| tail 3 | 0.75–1.01 | 0.95–1.00 | 1.05–1.49 | 1.00–1.01 | 0.87–1.04 | 0.86–1.03 | 0.83–1.00 | 1.79–1.84 | 1.04–1.11 | 1.83–1.87 |
| tail 4 | 0.74–1.00 | 0.95–1.00 | 0.98–1.33 | 1.00–1.00 | 0.86–1.03 | 0.85–1.01 | 0.84–1.00 | 1.17–1.21 | 0.93–1.07 | 1.15–1.20 |

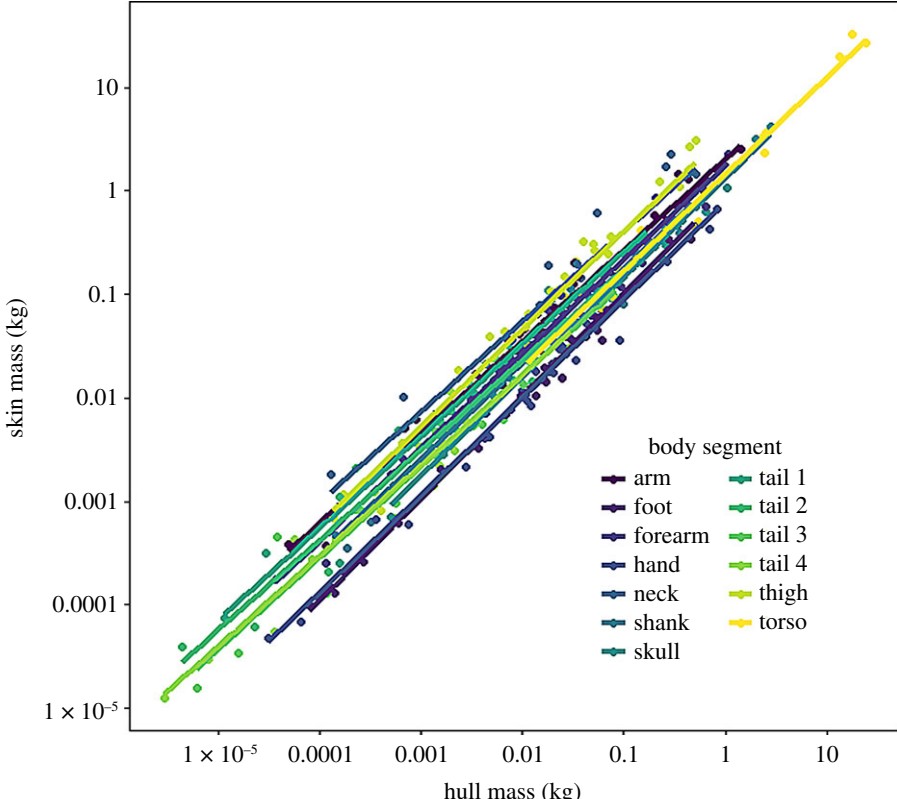

**Figure 3.** OLS regressions comparing the segmental volumes predicted by convex hull reconstruction and the actual values derived from the soft tissue segments. Logarithmic axes are displayed to aid comparison.

RMSE values were calculated for the OLS regressions using LOOCV and are reported alongside the regression coefficients. Overall, appendicular segments typically had slightly higher $R^2$ values than axial segments. The mean $R^2$ value for all appendicular segment–parameter relationships of 0.97 compared with a value of 0.93 for all axial comparisons. Among the appendicular segments, thighs were marginally less well predicted, while the neck and torso were the least predictable axial segments.

Convex hull reconstruction estimated segmental mass with a high degree of consistency (figure 3), with OLS regressions for mass having a mean $R^2$ value of 0.96. Reflecting the general trend across this study, $R^2$ values were higher in the appendicular segments. The position of the segmental centre of mass was extremely consistently reconstructed by convex hulls (figure 4), with a similar disparity between appendicular and axial segments. The slopes for centre of mass regressions were typically very close to one—with the major exceptions occurring in CM ($y$) regressions for axial segments. Across all segmental centre of mass regressions, the mean slope value was 0.99 and the mean $R^2$ was 0.94. The segmental moments of inertia in all three axes were similarly reliably predicted by convex hull reconstruction (figure 5), with a mean $R^2$ of 0.98. Considerably more variation was observed in the products of inertia regressions (figure 6), but $R^2$ was still consistently high—averaging 0.92.

## 4. Discussion

Assessing the results broadly, the reconstruction of appendicular segments tended to have more predictive power than in axial segments. This is probably in majority a by-product of the reduced sample size for axial segments, resulting from the limited availability of specimens with intact torsos or possessing tails. Even in the segments where convex hull reconstruction was least consistent, such as the neck, close relationships between convex hull and soft tissue segments were found for most BSPs. Those BSPs with less predictive relationships were typically products of inertia, which are commonly set to zero in MDA due to their often low values, as well as a desire for symmetry when modelling [13,34,50–52].

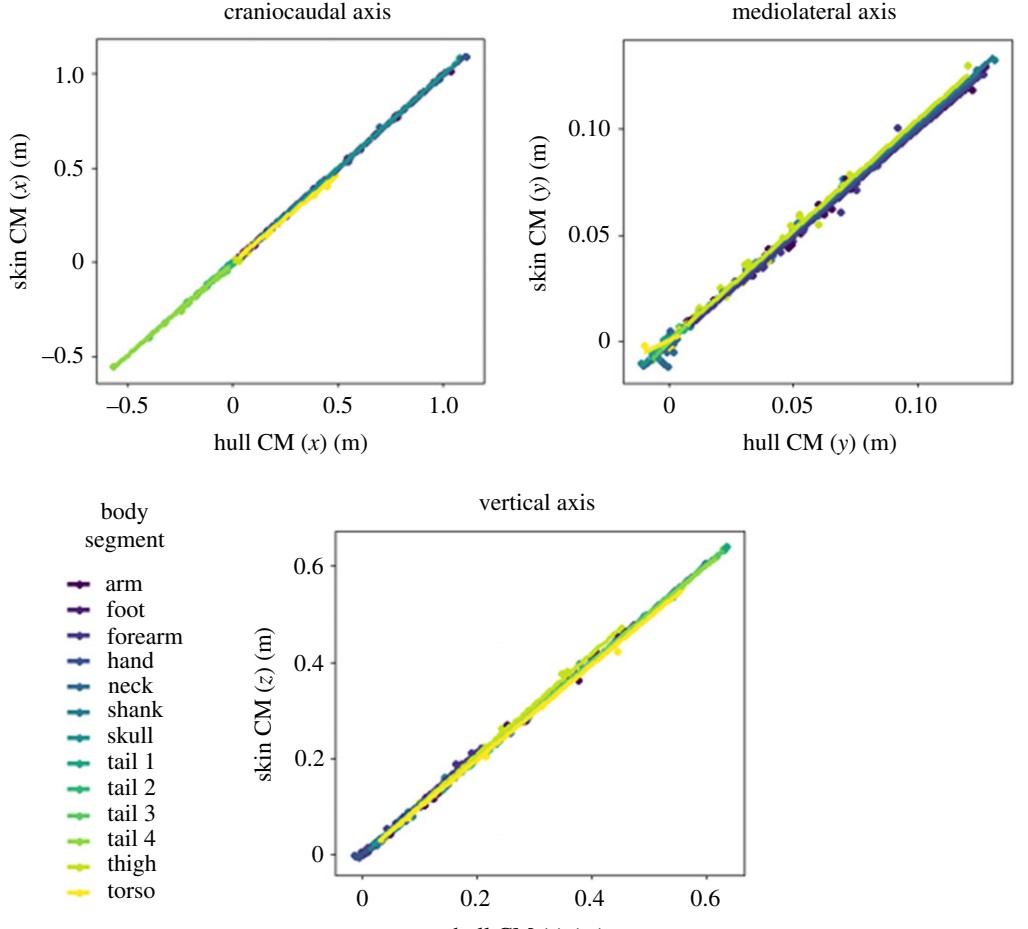

**Figure 4.** OLS regressions for the centre of mass coordinates in three axes: craniocaudal, mediolateral and vertical (as depicted in figure 2). In axial segments, CM ($y$) can be set to zero in MDA.

Convex hulls reconstructed segmental centre of mass extremely closely, as shown by the close grouping of slope values around one for centre of mass regressions. This indicates that the usage of convex hulls to estimate the segmental centre of mass for MDA is typically highly effective, even before the values are altered using the regression equations calculated here. As was noted previously, CM ($y$) can simply be set to zero in axial segments in MDA.

The thighs were the least consistently predictable appendicular segments, based on $R^2$ values—particularly in the products of inertia. The orientation of the specimens varied, making it more difficult to enforce an empirical measure for where the thigh segment ended and the torso began. In addition, bone comprises a lower proportion of the segmental mass in the thigh than in all other limb segments in humans—and presumably in other mammals [53]. Consequently, uncorrected convex hull reconstructions are likely to require more extensive conversion to replicate soft tissue segment values. This is borne out by the data, as naive convex hulls underestimated segmental mass by an average of 75%—the steepest underestimate among appendicular segments (and all segments other than the neck). Despite this, thigh regressions all showed highly significant correlations explaining most of the variance in the skin BSPs.

By contrast, the hands and feet were the only segments in which uncorrected convex hulls overestimated the segmental mass—a function of the skin wrapping more tightly around the underlying bone than in other segments [54]. Consequently, the high $R^2$ values in these segments could be expected. However, there was some variation in the orientation of the digits between the scanned specimens, as straightening the digits to standardize the orientation would have prevented accurate calculation of skin BSPs. The impact of this may be visible in the reduced effectiveness of CM ($z$) estimation—the position of the centre of mass in the vertical axis would be most affected by digits curling in the $z$-axis. Overall though, the strong predictive relationships are very encouraging for future reconstruction work.

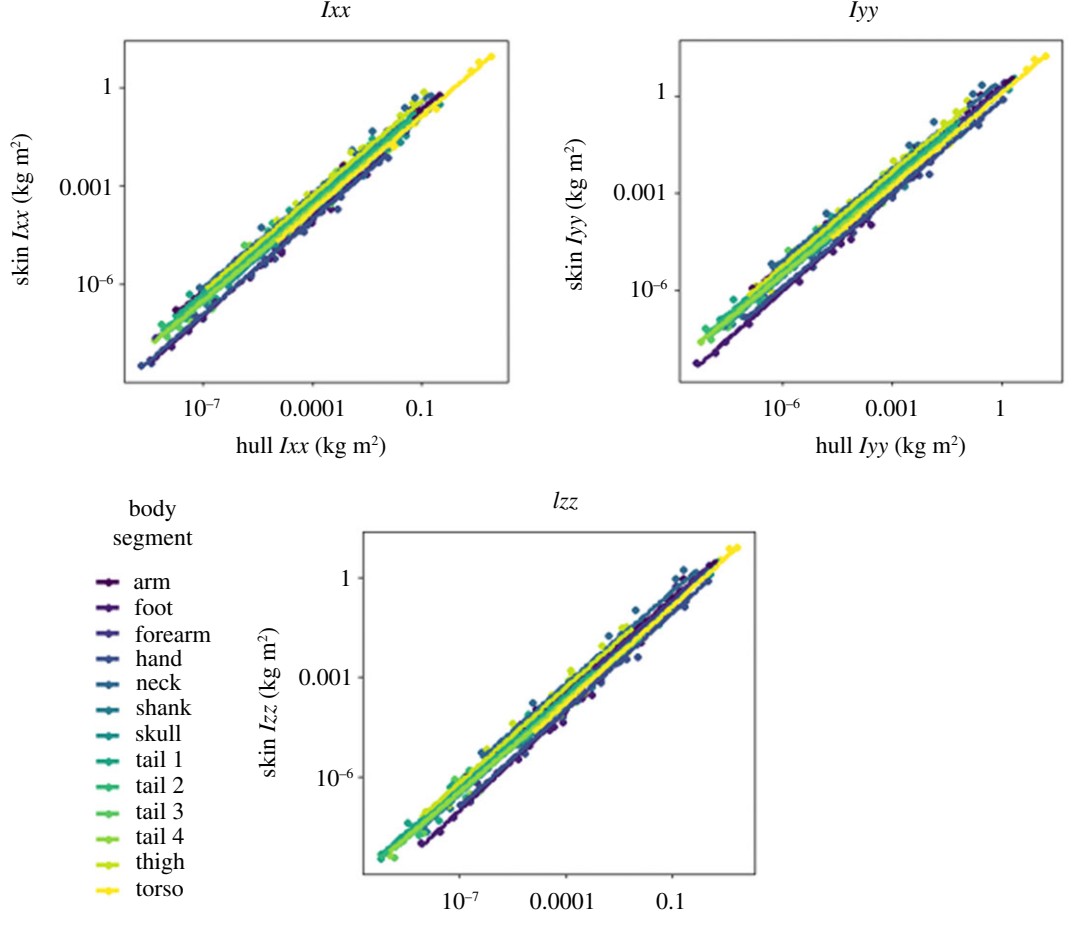

**Figure 5.** OLS regressions for the moments of inertia. Logarithmic axes are displayed to aid comparison.

In axial segments, the effectiveness of prediction was more variable, albeit typically still very effective. Much of this may be attributable to the reduced sample size, with $n$ ranging from 15 to 22 in axial segments, as opposed to 27–31 in appendicular segments. However, while a larger axial sample size would have been preferable, we still recommend these equations as the basis for convex hull reconstruction of mammals. BSPs of the head and tail were consistently predicted by convex hulls. Of all the segments investigated in this study, the BSPs of the neck were the least consistently predicted using convex hulls—although all regressions were highly significant with mostly high $R^2$ values. The notably broad confidence intervals for some parameters in the neck (tables 2 and 3) indicate that greater care should be taken when reconstructing necks. For reasons outlined above, torso sample size was considerably lower than in most segments ($n = 15$). Convex hull reconstruction of mammalian torsos was still mostly consistent—albeit with the second lowest mean $R^2$ value among segments. Caution is advised when reconstructing torsos: it is crucial to be mindful of the impact of the positioning and articulation of the specimen.

While BSPs are undoubtedly important in MDA [10], depending on the research question moderate variation in BSPs may have relatively little impact on the results [7,9]. Should this be the case, the greater efficiency with which convex hulls can be produced becomes a more significant factor in their favour. Not only does convex hull reconstruction estimate BSPs consistently, but it can do so in considerably less time without the need for anatomical expertise in a particular taxon or considerable proficiency with computer-aided design (CAD) software [2]. By providing a calibrating database of extant specimens this study greatly increases the utility of this approach for therian mammals.

Overall, convex hull reconstruction seems highly effective at estimating BSP values for use in multibody analyses of mammals. Substantial disparities in the resultant regression equations for different segments demonstrates the importance of treating each segment as distinct during reconstruction. The range of body sizes included in the dataset are worth keeping in mind when

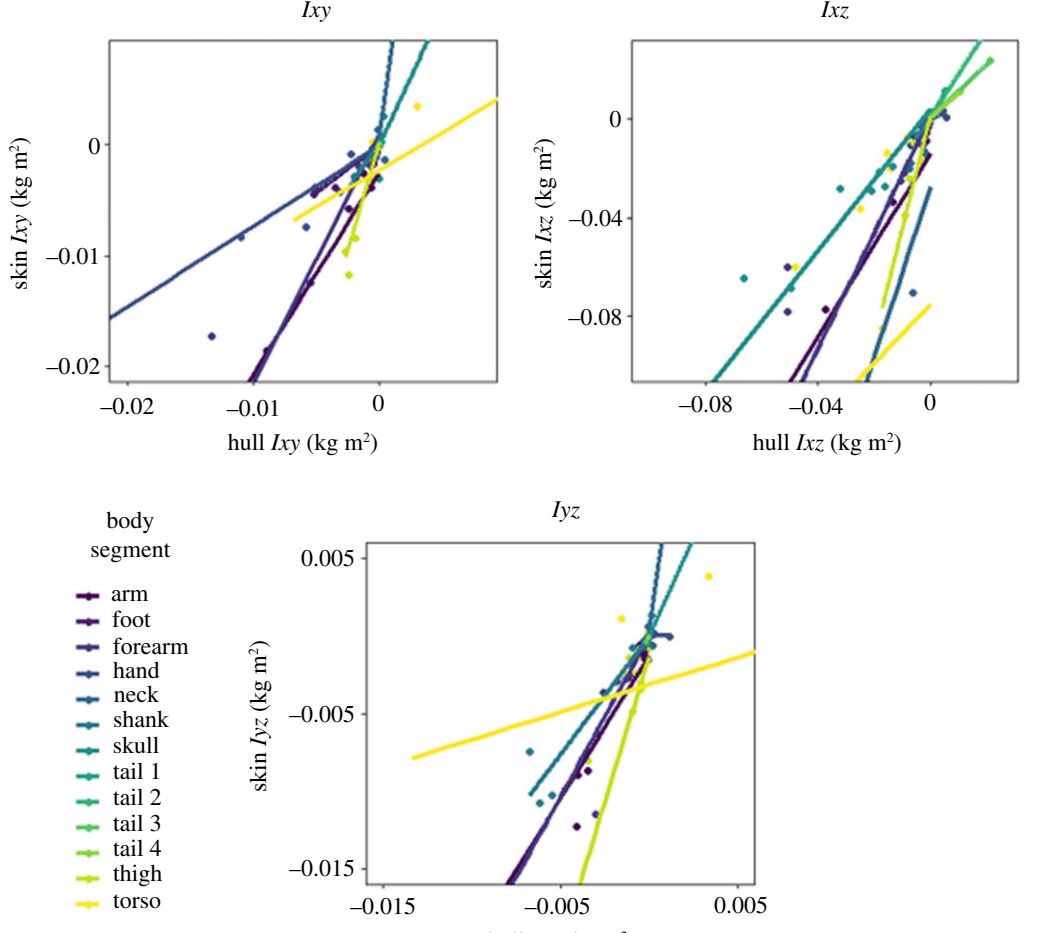

**Figure 6.** OLS regressions for the products of inertia. In axial segments, *Ixy* and *Iyz* can be set to zero in MDA.

planning future reconstructive research. The largest specimen is a polar bear (*U. maritimus*), which itself was missing multiple segments. It is important not to over-extrapolate values from the regression equations calculated here for species considerably larger than any included in producing the equations, although this may be inevitable for some extinct species [24,55].

# 5. Conclusion

This study uses a validatory dataset of extant mammals to produce predictive equations for usage with convex hull reconstruction of mammals. Regressions revealed strong, significant predictive relationships between BSPs estimated using convex hulls and those calculated from the corresponding soft tissue segments of a wide variety of mammal species. The resultant regression equations are made available for use, providing a quick, validated method for BSP estimation in mammals. These can be used for volumetric reconstruction as well as further biomechanical analysis, with both extinct mammal species and extant species for which full body scan acquisition is difficult.

Ethics. This article does not present research with ethical considerations. No live animals were used by the authors in the study.

Data accessibility. All datasets supporting this article have been uploaded to the Dryad Digital Repository (https://doi.org/10.5061/dryad.cfxpnvx4k) [56]. The data are provided in electronic supplementary material [57].

Authors' contributions. S.J.C., W.I.S. and T.A.P. designed the study. S.J.C. and T.A.P. carried out the analysis. S.J.C. processed all scans used in the study and wrote the manuscript, with revision from all authors. Final approval for the paper was given by all authors.

Competing interests. We declare that we have no competing interests.

Funding. S.J.C. was funded by the Manchester Environmental Research Institute. W.I.S. was funded by the National Environmental Research Council (grant no. NE/R011168/1). T.A.P. was funded by the Leverhulme Early Career Fellowship (grant no. ECF-2018-264).

Acknowledgements. We would like to thank the following people and organizations for providing the scans for this paper and assisting with its conception: Dr Naomi Wada, Dr Karl Bates, Dr Charlotte Brassey, Prof. Shin-Ichi Fujiwara, the Kyoto University Primate Research Institute, MXR Imaging and Santa Barbara Zoo.

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
