## [Peer Review File · Royal Society Open Science]

Review History

Decision letter (RSOS-210836.R0)

Dear Mr Coatham:

I am pleased to inform you that your manuscript entitled "Convex hull estimation of mammalian body segment parameters" is now accepted for publication in Royal Society Open Science.

You can expect to receive a proof of your article in the near future. Please contact the editorial office (openscience@royalsociety.org) and the production office (openscience_proofs@royalsociety.org) to let us know if you are likely to be away from e-mail contact -- if you are going to be away, please nominate a co-author (if available) to manage the proofing process, and ensure they are copied into your email to the journal. Due to rapid

publication and an extremely tight schedule, if comments are not received, your paper may experience a delay in publication.

on behalf of Dr Jonas Rubenson (Associate Editor) and Professor Kevin Padian (Subject Editor).

Associate Editor (Dr Jonas Rubenson) Comments to Author:

Dear Dr. Coatham,

I am pleased to see that in your revised manuscript you and your co-authors have addressed the previous reviewer comments (J. R. Soc. Interface) well. The previous reviewer's indicate that your study is sound, and that it provides important new data and analyses. I agree. Thank you for your submission to RSOS!
